# Targeting Angiogenesis by Blocking the ATM–SerRS–VEGFA Pathway for UV-Induced Skin Photodamage and Melanoma Growth

**DOI:** 10.3390/cancers11121847

**Published:** 2019-11-22

**Authors:** Yadong Song, Hongyan Lu, Qiong Wang, Rong Xiang

**Affiliations:** School of Medicine, Nankai University, Tianjin 300071, China; 1120160479@mail.nankai.edu.cn (Y.S.); 1120160482@mail.nankai.edu.cn (H.L.); trancea@163.com (Q.W.)

**Keywords:** UV, VEGFA, ATM, all-trans retinoic acid (*t*RA), seryl tRNA synthetase (SerRS), KU55933, melanoma, photodamage

## Abstract

Retinoic acid (RA) has been widely used to protect skin from photo damage and skin carcinomas caused by solar ultraviolet (UV) irradiation, yet the mechanism remains elusive. Here, we report that all-trans retinoic acid (*t*RA) can directly induce the expression of a newly identified potent anti-angiogenic factor, seryl tRNA synthetase (SerRS), whose angiostatic role can, however, be inhibited by UV-activated ataxia telangiectasia mutated (ATM) kinase. In both a human epidermal cell line, HaCaT, and a mouse melanoma B16F10 cell line, we found that *t*RA could activate SerRS transcription through binding with the SerRS promoter. However, UV irradiation induced activation of ATM-phosphorylated SerRS, leading to the inactivation of SerRS as a transcriptional repressor of vascular endothelial growth factor A (VEGFA), which dampened the effect of *t*RA. When combined with ATM inhibitor KU-55933, *t*RA showed a greatly enhanced efficiency in inhibiting VEGFA expression and a much better protection of mouse skin from photo damage. Also, we found the combination greatly inhibited tumor angiogenesis and growth in mouse melanoma xenograft in vivo. Taken together, *t*RA combined with an ATM inhibitor can greatly enhance the anti-angiogenic activity of SerRS under UV irradiation and could be a better strategy in protecting skin from angiogenesis-associated skin damage and melanoma caused by UV radiation.

## 1. Introduction

Excessive exposure to sunlight causes sunburn, primarily because the short wavelength (280–320 nm) ultraviolet (UV)-B irradiation causes skin damage [1]. Acute sunburn or photo damage causes injury to both epidermal and dermal layers and is characterized by erythema, edema, and tenderness. Though the damage is often local and controllable, severe sunburn on a large portion of the body surface area may lead to systemic effects, such as fever and prostration. Chronic unprotected exposure to UV irradiation is considered to be an important cause of carcinogenesis on the skin [2,3,4].

The effects of various treatments for acute photo damage have been assessed over the years, including topical ointments, oral pills, and laser and surgical procedures. Retinoids are a class of compounds derived from vitamin A or vitamin A analogs, which are frequently recommended by dermatologists and widely used on sunburned patients [5,6,7]. All-trans retinoic acid (*t*RA) is the first generation of retinoids. It has been adopted in clinical use for several decades and still is popular for its low cost. Retinoic effects on photo-damaged skin include improvement of coarse wrinkling and dyschromia, as well as decreased roughness and vascularization. However, the efficiency of monotherapy is limited; furthermore, the toxicity and teratogenicity would arise along with the increase of concentration and extension of treatment period [8]. Therefore, combined applications of *t*RA and drugs with similar effects but different mechanisms are urgently needed.

UV-induced imbalance between angiogenic and angiostatic factors is associated with many skin pathologies including aging and carcinogenesis [9,10,11,12]. *t*RA has been reported to be able to inhibit the expression of vascular endothelial growth factor (VEGF) and angiogenesis in various tumors and human skin [9,13,14]. VEGFA is the most extensively studied among all members of the VEGF family, which includes VEGFB, VEGFC, and placental growth factor (PIGF), for its key role in regulating angiogenesis [15,16,17]. Improper angiogenesis is an obvious characteristic in tumor development because tumors need an abundant nutrient supply in order to support the rapid proliferation of cancer cells [18,19]. Angiogenesis has been validated as an ideal target in tumor therapy for more than a decade, and inhibitors against the VEGF pathway are among the most promising anti-tumor reagents [20,21,22].

Here, we report that *t*RA can promote the expression of a newly identified, potent anti-angiogenic factor, namely, seryl-tRNA synthetase (SerRS), by triggering the binding of a retinoic acid receptor (RAR) on the SerRS promoter. As a member of aminoacyl tRNA synthetase that functions in protein biosynthesis, SerRS in vertebrates has a unique domain that contains a nuclear localization signal, which directs a portion of SerRS into the nucleus, where it directly binds on the VEGFA promoter to suppress VEGFA transcription [23,24]. However, our previous study demonstrated that hypoxia-activated ataxia telangiectasia mutated (ATM)/ATR kinases can phosphorylate SerRS at serine 101 (S101) and serine 214 (S241) residues, leading to decreased binding of SerRS on the VEGFA promoter and activated VEGFA expression. Therefore, we hypothesized that *t*RA might prevent skin angiogenesis-associated skin damage by activating SerRS expression, which could then be inactivated by UV-induced ATM activation, leading to a dampened effect of *t*RA. Therefore, *t*RA combined with the ATM inhibitor might improve the efficiency of protecting skin from angiogenesis-associated pathologies, including skin aging and skin tumors.

## 2. Results

### 2.1. The ATM–SerRS–VEGFA Pathway Played an Essential Role in UV-Induced VEGFA Expression in HaCaT Cells

To confirm the effects of UV on the induction of *VEGFA* in human skin cells, HaCaT cells, a human keratinocyte cell line, were irradiated with UV. *VEGFA* induction reached the highest level (approximately fourfold) at 5 h post-irradiation (Figure 1a). We also tested different dosages of UV irradiation for *VEGFA* induction, and we found that 420 J/m^2^ of UV irradiation achieved the highest *VEGFA* induction (Figure 1b). A higher dosage of UV irradiation triggered cell death, resulting in less *VEGFA* induction (Figure 1b). Consistently, secreted *VEGFA* protein was further confirmed by ELISA at different time points (Figure 1c). Secreted *VEGFA* protein levels were significantly increased by approximately fivefold at 5 h post-UV treatment and kept increasing until 9 h post UV treatment (Figure 1c).

In previous studies, SerRS was identified to be a potent transcriptional repressor of *VEGFA* [24], and hypoxia-induced activation of ATM/ATR could phosphorylate SerRS at S101 and S241 residues to release SerRS from *VEGFA* promoter, greatly contributing to hypoxia-induced angiogenesis. It has been well established that ATM is a core kinase in the UV-induced DNA repair response. We hypothesized that UV-induced activation of ATM might phosphorylate and inactivate SerRS to increase *VEGFA* transcription. To confirm this hypothesis, we first treated HaCaT cells with UV irradiation. Activation of ATM was observed to quickly peak a half-hour after treatment, followed by the phosphorylation of SerRS, which peaked at 1 h post treatment and maintained a relatively lower level until 5 h post treatment (Figure 1d).

To gain additional insights into the role of ATM activation in UV-induced *VEGFA*, we utilized a specific ATM inhibitor (ATMi), KU-55933, to block ATM activation during UV irradiation (Figure 1e), where SerRS phosphorylation was also greatly inhibited (Figure 1e). HaCaT cells pretreated with KU-55933 showed no *VEGFA* induction at all upon UV irradiation (Figure 1f,g), strongly indicating that ATM activation is necessary for UV-induced *VEGFA* expression.

To further demonstrate whether the phosphorylation of SerRS is essential for the induction of *VEGFA* by UV, we mutated S101 and S241 of SerRS to alanine (i.e., SerRS^AA^) or to aspartic acid residues (SerRS^DD^) to mimic the phosphorylation-deficient and phosphorylated SerRS forms, respectively. These mutants were stably transfected into HaCaT cells (Figure 1h). UV irradiation could still induce *VEGFA* expression in wild-type SerRS (SerRS^WT^) or SerRS^DD^-transfected HaCaT cells, although to a lesser degree when compared with empty vector transfected cells (Figure 1i,j). However, overexpression of phosphorylation-deficient SerRS^AA^ completely inhibited UV-induced *VEGFA* expression in HaCaT cells (Figure 1i,j), suggesting that SerRS phosphorylation by UV-activated ATM was essential for *VEGFA* induction.

### 2.2. tRA Enhanced the Transcription of SerRS to Suppress VEGFA Expression in HaCaT Cells

*t*RA has been reported to be able to inhibit UV-induced *VEGFA* expression and skin angiogenesis [9]. We first confirmed that, in HaCaT cells, *t*RA was able to reduce UV-induced *VEGFA* expression in a dose-dependent manner (Figure 2a,b). Similar effects were also achieved by the overexpression of SerRS (Figure 1i,j). To investigate if *t*RA inhibited *VEGFA* induction through regulating SerRS expression, we first searched the SerRS promoter for possible binding sites of the *t*RA receptor, namely, retinoic acid receptors (RARs) [8]. We found two candidate retinoic acid response elements (RAREs) at ‘(−2362) 5’-GGAACACCTGAGGTCA-3’ (−2346)’ (RARE1) and ‘(−874) 5’-AGATCAAGAAGGTCT-3’ (−859)’ (RARE2), which harbor the well-established RAR-binding motif ‘(A/G)G(T/G)TCA(Xn)(A/G)G(T/G)TCA’ [25] (Figure 2c). To test if the SerRS promoter was regulated by *t*RA, we ligated the SerRS promoter to the luciferase reporter gene (Figure 2c). As expected, *t*RA could enhance the expression of the reporter gene driven by the SerRS promoter (Figure 2d). Deletion of either RARE1 or RARE2 could completely abolish *t*RA-enhanced expression of the reporter gene (Figure 2e), confirming that these two sites on the SerRS promoter were true RAR-binding sites.

To investigate if *t*RA could enhance the expression of endogenous SerRS, we treated HaCaT cells with different dosages of *t*RA and observed increased mRNA and protein levels of SerRS (Figure 2f,g). Taken together, our results suggested that *t*RA might enhance SerRS expression to repress *VEGFA* transcription.

### 2.3. ATM Inhibitor Enhanced the Effect of tRA on the Protection of Mouse Skin from UV-Induced VEGFA Expression

Given that the application of *t*RA alone did not fully inhibit UV-induced VEGFA (Figure 2a,b), while overexpression of the phosphorylation-deficient SerRS mutant could almost fully repress UV-induced VEGFA (Figure 1i,j), we suspected that the inefficiency of *t*RA might be due to SerRS phosphorylation by UV-activated ATM, which inhibited the transcriptional repressor activity of SerRS. Thus, we hypothesized that the ATM inhibitor should be able to increase the effect of *t*RA against UV-induced VEGFA. To test this, we treated mice with *t*RA alone or *t*RA combined with ATM inhibitor KU-55933 before acute UV irradiation on the mouse skin (Figure 3a). As reported, UV irradiation caused a dramatic increase in epidermal VEGFA mRNA and protein levels, which could be prevented by *t*RA alone (Figure 3b,c). Consistent with the in vitro results in cultured HaCaT cells (Figure 1f,g), the ATM inhibitor alone could also prevent UV-induced VEGFA in mouse skin (Figure 3b,c). As expected, the combined application of *t*RA and ATM inhibitor fully inhibited UV-induced VEGFA expression in mouse skin (Figure 3b,c), further supporting that the UV–ATM–SerRS pathway plays an essential role in the regulation of cutaneous angiogenesis. These results were further confirmed by immunohistochemistry (IHC) (Figure 3d,e) and immunofluorescent staining (Figure 3f,g).

### 2.4. The ATM Inhibitor Improved the Efficiency of tRA Against Pathological Skin Angiogenesis Caused by UV Irradiation in Mice

We also investigated whether pretreatment with *t*RA and ATMi could inhibit acute UV irradiation-induced skin angiogenesis. Mice were treated as described in “Materials and Methods”. Skin samples were obtained 5 h after the third round of UV radiation, and skin vascularization was analyzed by H&E staining and immunohistochemistry of platelet/endothelial cell adhesion molecule-1 (CD31, also known as PECAM1) (Figure 4a). Acute UV exposure dramatically increased the vascularity of the dermis; the average size of vessels enlarged up to threefold and the number increased up to 1.5-fold compared to mice without irradiation. Topical application with *t*RA or intraperitoneal injection with ATMi significantly reduced the vascularity caused by UV exposure. The increases in the size and the number of dermal blood vessels were both inhibited, especially when treated with *t*RA and ATMi together (Figure 4b–d).

### 2.5. ATM Inhibitor Improved the Effects of tRA on Alleviating UVB-Induced Thickening and the Inflammatory Response of the Epidermis

UV-induced photo damage results in various pathologic alterations on the skin, such as epidermal thickness, skin hydration, erythema, wrinkle formation, and collagen degradation. Photos of the bare skin were taken every day to record the changes, and Masson staining was adopted to demonstrate the thickness of the epidermis and the morphological characteristics of collagen (Figure 5a,b). The Local Skin Response Grading Scale (LSR) was also applied to quantify the photo damage on skin (Figure 5c and Appendix A
Table A1). Acute UV exposure led to obvious skin erythema, coarseness, thickening, edema, and the appearance of scales in mice by day 2 after UV irradiation. Treatments with *t*RA and ATMi together achieved the best skin protection against these pathological changes when compared to groups treated with Vaseline control, *t*RA alone, or ATMi alone (Figure 5a–c).

### 2.6. tRA and ATMi Could Greatly Inhibit UV-Induced VEGFA in Melanoma Cells

To gain a better understanding of the biological significance of the UV–ATM–SerRS pathway in skin cancer, we further investigated the activation of this pathway in the skin cancer melanoma. We used a mouse melanoma cell line, B16F10, to analyze the role of the UV–ATM–SerRS pathway in regulating *VEGFA* and tumor angiogenesis. Compared with HaCaT cells, B16F10 seemed more sensitive to UV radiation, in which 90 J/m^2^ of UV irradiation was the highest induction reached for *VEGFA* within a similar period of time after radiation (Figure 6a,b). ATM was phosphorylated rapidly half an hour after UV irradiation, which was followed by SerRS phosphorylation soon after (Figure 6c). Inhibition of ATM by KU-55933 in B16F10 strongly reduced the phosphorylation of SerRS (Figure 6d) and *VEGFA* induction (Figure 6e,f) by UV treatment, suggesting ATM activation was a key step in UV-induced angiogenesis.

We also stably transfected B16F10 cells with SerRS^WT^, SerRS^AA^, and SerRS^DD^ (Figure 6g). Consistent with the results in HaCaT, overexpression of SerRS^WT^ could only attenuate *VEGFA* induction by UV, whereas phosphorylation-deficient SerRS^AA^ almost fully inhibited UV-induced *VEGFA* expression (Figure 6h,i), highlighting that UV-induced SerRS phosphorylation by ATM was important for *VEGFA* induction. *t*RA was also able to promote SerRS expression (Figure 6j) and inhibit UV-induced *VEGFA* expression in B16F10 cells (Figure 6k,l). These results suggested that the ATM–SerRS–VEGFA pathway is important for UV-induced angiogenesis in melanoma.

### 2.7. The ATM Inhibitor Greatly Improved the Effect of tRA in Inhibiting the Growth of a Melanoma Xenograft in Mice

Chronic UV irradiation has been reported to be able to promote the progression of malignant skin melanoma [26,27]. These findings are consistent with our results showing the role of UV in the induction of *VEGFA* in melanoma cells, which could promote tumor angiogenesis and growth. Therefore, we hypothesized that the combined application of *t*RA with the ATM inhibitor might greatly inhibit the progression of melanoma by targeting the UV–ATM–SerRS–VEGFA signaling pathway to suppress tumor angiogenesis. To test this hypothesis, we established a melanoma xenograft model by inoculating B16F10 cells subcutaneously into mice and treating them with UV and *t*RA/KU-95533, as mentioned in the Materials and Methods section (Figure 7a). As reported, UV irradiation greatly increased melanoma growth (Figure 7b,c). The combined application of *t*RA and ATM inhibitor dramatically inhibited melanoma growth whether the xenografts were UV-irradiated or not (Figure 7b,c), which was not due to the toxicity of the reagents as the body weights of all the mice were not that different (Figure 7d). As expected, application of *t*RA together with the ATM inhibitor greatly reduced the *VEGFA* expression in melanoma xenografts (Figure 7e,f).

## 3. Discussion

Because the skin is the largest organ of the human body and is almost completely exposed to the external environment, it is therefore more susceptible to external environmental influences and damage than other organs. Excessive UV radiation from sunlight is the most dangerous element for cutaneous photodamage, photoaging, and tumors [28,29,30]. The characteristic feature among various pathologic alterations caused by UV exposure is the activation of angiogenesis, which is supposed to be responsible for tumorigenesis and tumor growth.

Anti-angiogenic therapies have produced promising results for multiple malignancies over recent decades. Among these therapies, bevacizumab is the best-known monoclonal antibody that neutralizes VEGFA, a key pro-angiogenic factor [31,32,33]. Bevacizumab is approved by the Food and Drug Administration (FDA) for various cancer therapies, including non-small cell lung cancer, metastatic colorectal cancer, and recurrent glioblastoma [34,35,36,37]. Almost all the anti-VEGF therapeutic approaches present are monoclonal antibodies directly targeting one or more mature isoforms of VEGF in serum, or VEGF receptors, thereby inhibiting tumor growth [18,38]. Such limited mechanisms result in the urgent issue of understanding why the majority of patients do not respond or stop responding to such drugs. Therefore, our study on the regulation of VEGFA transcription—the source of this important growth factor—is innovative and important.

Several independent forward studies have suggested an essential role of SerRS in vascular development [23,24,39,40,41]. SerRS is well-known for its essential function in aminoacylation of tRNA^Ser^ for protein synthesis in the cytoplasm. However, the role of SerRS in vascular development is independent from its enzymatic activity, and this role is specific to SerRS among all tRNA synthetases. As a transcriptional repressor of VEGFA, SerRS directly binds to its promoter, recruiting histone deacetylase to condense the chromatin at the promoter region of VEGFA and thereby shutting down gene transcription.

Our previous study showed that hypoxia, a common feature of solid tumors, could induce VEGFA expression and angiogenesis in breast cancer cells and tumor-bearing mice by activating the ATM–SerRS pathway. Hypoxia frequently activates DNA damage and repair pathways, including the phosphorylation of ATM at Ser1981. Phosphorylated ATM resulted in the phosphorylation of downstream SerRS, decreasing the affinity of SerRS with DNA, and the inhibitory effect on VEGFA was weakened. Enlightened by this discovery, and on the basis of knowledge that UV radiation could also induce the phosphorylation of ATM, we hypothesized that the ATM–SerRS pathway might play a role in UV-induced VEGFA overexpression as well [42,43,44].

In this study, we verified in vitro and in vivo that UV radiation induced VEGFA overexpression through the ATM–SerRS pathway. When we irradiated HaCaT cells or mice with UV, VEGFA was overexpressed and was accompanied by the phosphorylation of ATM and SerRS. When KU-55933, a specific ATM inhibitor, was applied, the overexpression of VEGFA was reversed, and the phosphorylation of ATM and SerRS was suppressed simultaneously. In fact, ATM was reported to be necessary for angiogenesis in previous investigations through different pathways [45,46]. Our findings consider a new interpretation for the usage of an ATM inhibitor for the therapy of pathological angiogenesis caused by UV exposure.

*t*RA is another key weapon we acquired. It is well known that *t*RA is a differentiating agent used in the therapy of acute promyelocytic leukemia (APL) therapy, and it is under-used in other malignancies considering its low cost and low systemic toxicity [47,48,49]. In theory, *t*RA is an ideal candidate for cancer chemoprevention, as cancer is commonly characterized by abnormal growth with a lack of differentiation, which could be modified by *t*RA, and the combined applications of *t*RA with other drugs are also in clinical trials [50,51,52]. However, the knowledge of *t*RA inhibiting angiogenesis is rare, and this limits its usage in cancer therapy. We analyzed the promoter region of SerRS and determined two binding sites for a *t*RA receptor, named RARE1 and RARE2, which were verified with western blotting and a dual-luciferase reporter assay, as shown above. Our data indicated that *t*RA bonds to the promoter of SerRS and exerts its inhibitory efficiency on the expression of VEGFA by promoting the transcription of SerRS.

Immunotherapy with PD-L1/PD-1 inhibitors has demonstrated clinical benefit across a wide range of cancer types, especially malignant melanoma. However, patients who experience durable response and survival are only a limit subset of those treated. Therefore, we need to identify novel combinations. In fact, the biologic role of VEGFA has extended beyond its impact on angiogenesis over the years. It can have a direct effect on multiple cells involved in immunity, including T cells, dendritic cells, regulatory T cells, and myeloid-derived suppressor cells [53]. Recent studies have also showed that anti-VEGF therapy can improve anti-PD-L1 (programmed death ligand 1) treatment, especially when it generates intratumoral high endothelial venules (HEVs) that facilitate enhanced cytotoxic T lymphocyte (CTL) infiltration and tumor cell destruction [54]. Recent clinical studies also support the role of VEGFA in anti-cancer immunity [55,56].

In spite of these new findings, human application of our achievement needs additional research, especially in melanoma patients. First, we need to further confirm the safety and efficiency by combining the ATM inhibitor and *t*RA in humans. Problems related to the doses, applied methods, and the carriers should be considered preferentially. Second, though it is tempting to speculate the anti-vascular effects of VEGF inhibition in combination with immunotherapy, the mechanism of the anti-VEGF/anti-PD-L1 interaction is multifactorial [57,58]. Third, toxicity from those immuno-checkpoint inhibitors already occurs frequently, especially neurotoxicity and cardiotoxicity, where rates of moderate and severe immune-related adverse events range from 10% to 55% [57,58], and the addition of *t*RA and ATM inhibitor may increase such toxicity further. However, the ability to overcome these difficulties is still promising and anticipated.

## 4. Materials and Methods

### 4.1. Regents and Antibodies

Cell culture medium, DMEM, and fetal bovine serum were purchased from Thermo Fisher Scientific (Waltham, MA, USA). *t*RA (S1653) and ATM inhibitor (KU-55933) were from Selleck Chemicals (Houston, TX, USA). For RT-PCR, SerRS, VEGFA, β-actin, IL1b, IL10, TNF-α, INF-α, and GAPDH primer sets were synthesized by Sangon Biotech (Shanghai, China). For immunohistochemistry, immunofluorescence, and western blotting, anti-mouse CD31 (ab28364) and VEGFA (ab52917) antibodies were purchased from Abcam (Cambridge, UK). Anti-human VEGFA was from Proteintech Group (19003-1-AP, Wuhan, China), and anti-mouse SerRS was from Sigma-Aldrich (WH0006301M1, Darmstadt, Germany). Anti-P-ATM (4526), anti-ATM (2873), and anti-phosphor-ATM/ATR Substrate (S*Q) (9607) were from Cell Signaling Technology (Beverly, MA, USA). For ELISA, VEGFA kits (ab119566, ab119565) were purchased from Abcam (Cambridge, UK).

### 4.2. Cell Culture and UV Exposure

The human epidermal cell line HaCaT and mouse melanoma cell line B16F10 were obtained from Procell Life Science and Technology (Wuhan, China). They were cultured in DMEM, supplemented with 10% fetal bovine serum and 1% penicillin–streptomycin, and grown on plastic tissue culture dishes in a humidified 5% CO_2_ atmosphere at 37 °C. The UVB source was produced by an ultraviolet crosslinker (UVP CL-1000M, Thermo Fisher Scientific, Waltham, MA, USA), which supplied a uniform intensity of UVB exposure at a constant emission spectrum of 302 nm. Cells were seeded at 5 × 10^5^ cells/well on 60 mm-well plates and irradiated with UV 24 h later. Subconfluent monolayer cells were rinsed with phosphate-buffered saline (PBS) before radiation. Sham-irradiated control cells were handled in an identical manner, but they were not exposed to UVB. After irradiation, medium was replenished with fresh DMEM, and cells were cultured for various times. In experiments involving treatment with ATMi, cells were pretreated 2 hours before UVB irradiation. In experiments involving *t*RA, cells were pre-incubated with *t*RA of different concentrations for 24 h. All reagents were dissolved in DMSO, and cells were treated with the same amounts of DMSO in medium before radiation.

### 4.3. Animals and UVB Exposure

C57BL/6 mice were purchased from Vital River Laboratory Animal Technology Company (Beijing, China) and maintained in a specific pathogen-free facility. Animal use complied with Nankai University Animal Welfare Guidelines and were approved by the Nankai University Animal Care and Use Committee (Ethic approved number: 20180004). Hair on the dorsal side was shaved and removed, depilatory cream (VEET, RB Group of Companies, Berkshire, UK) was also used to clean the back thoroughly. For tumor transplantation, a total of 2 × 10^5^ cells were injected intracutaneously into the right flanks. *t*RA was smeared on the shaved back, and KU-55933 was intraperitoneally injected one hour before UVB exposure. Vaseline (Unilever, London, UK) was smeared as a control. Mice were fixed to be completely bare on the back during irradiation. Eight mice were in each group, and a single dose of UVB was 160 J/m^2^ at a frequency of once every two days, with the total dose of 480 J/m^2^. Skin responses were recorded daily and quantified on the basis of the LSR Grading Scale every day, including erythema, scales, edema, and roughness.

### 4.4. RNA Extraction and RT-PCR

HaCaT, B16F10 cells, or skin samples were collected, and total RNA was isolated with Trizol (Invitrogen, Waltham, MA, USA) and purified according to the protocol. Reverse transcription was performed with the Superscript Ⅱ system and Oligo-dT primers (Invitrogen). Real-time PCR analysis was performed with diluted cDNA and Fast SYBR Green Master Mix (Applied Biosystems, Waltham, MA, USA) using a LightCycler96 real-time PCR system (Roche Life Science, Basel, Switzerland). Relative expression to the reference gene β-actin was calculated with the Ct method using the following equations:

△Ct (Sample) = Ct (target) – Ct (reference); relative quantity = 2^−△Ct^.

The following primers were used for the quantification of gene expression: hACTB-F, 5’-CGTCACCAACTGGGACGA-3’ and hACTB-R, 5’-ATGGGGGAGGGCATACC-3’; hVEGFA-F, 5’-GAGGGCAGAATCATCACGAAG-3’ and hVEGFA-R, 5’- TGTGCTGTAGGAAGCTCATCTCTC-3’; mACTB-F, 5’-GGCTGTATTCCCCTCCATCG-3’ and mACTB-R, 5’-GCACAGGGTGCTCCTCAG-3’; mVEGFA-F, 5’-GTCCGATTGAGACCCTGGTG-3’ and mVEGFA-R, 5’-TTGACCCTTTCCCTTTCCTCG-3’; mINF-⍺-F, 5’-GTGGTTTGCTACGACGTGGG-3’ and mINF-⍺-R, 5’-ATGACTGTGCCGTGGCAGTA-3’; mIL1b-F, 5’-CAACCAACAAGTGATATTCTCCATG-3’ and mIL1b-R, 5’-GATCCACACTCTCCAGCTGCA-3’; mIL10-F, 5’-TTTGAATTCCCTGGGTGAGAA-3’ and mIL10-R, 5’-CTCCACTGCCTTGCTCTTATTTTC-3’; mTNF-⍺-F, 5’-TTCTGTCTACTGAACTTCGGGGTGATCGGTCC-3’ and mTNF-⍺-R, 5’-GTATGAGATAGCAAATCGGCTGACGGTGTGGG-3’.

### 4.5. Co-Immunoprecipitation Assay and Western Blotting Analysis

HaCaT cells were resuspended on ice with lysis buffer (20 mM Tris-HCl [pH 7.5], 150 mM NaCl, 1 mM EDTA, 1 mM EGTA, 1% Triton X-100, 2.5 mM sodium pyrophosphate, 1 mM beta-glycerophosphate, 1 mM Na_3_VO_4_, and protease inhibitor cocktail). Supernatants were incubated with indicated antibodies and protein-G-conjugated agarose beads (Invitrogen) for at least 1 h. The beads were washed three times with wash buffer (Triton X-100 was reduced from 1% to 0.1% compared with lysis buffer) and then was subjected to SDS-PAGE and western blotting analysis with indicated antibodies.

### 4.6. Measurement of VEGFA Secretion

Cell-free supernatants were collected after UVB irradiation and stored at −80 °C until required for the cytokine assay. Secreted *VEGFA* levels in the culture media of HaCaT cells were determined by sandwich ELISA, according to the manufacturer’s protocol. Absorption of the avidin-horseradish peroxidase color reaction was measured at 405 nm, and VEGFA concentrations were determined using a standard graph prepared using serial dilutions of human recombinant human VEGFA as a standard.

### 4.7. H&E, Immunohistochemistry, and Immunofluorescent Staining

After euthanasia, biopsies from the UV-radiated dorsal skin or tumor tissue samples of the mice were captured, fixed in 4% formalin, embedded in paraffin, and cut into 5 μm slices. For HE staining, slices were stained with hematoxylin and eosin. Immunohistochemical analysis was performed using an immunohistochemistry (IHC) kit, according to the manufacturer’s instructions. In brief, slices were deparaffinized in xylene and rehydrated through a graded ethanol series. The primary antibodies used for immunohistochemistry were mouse anti-CD31 (1:200 dilution, Abcam) and mouse anti-VEGFA (1:200 dilution, Proteintech Group). The numbers and diameters of cutaneous blood vessels were captured and analyzed using an image analysis program (Image-Pro Plus, Olympus, Fukuoka, Japan). Vessel numbers per mm^2^, average vessel sizes, and the relative areas occupied by vessels were determined in the dermis. Three different fields from the epidermal–dermal junction were examined in the dermis of each section. Measurements of vessel number, vessel size, collagen and epidermis thickness, and CD31-positive cells were performed in a blinded manner. Immunofluorescence of VEGFA was detected using mouse anti-VEGFA (1:200 dilution, Proteintech Group), and the nuclei were stained with DAPI (4’,6-diamidino-2-phenylindole) (Sigma-Aldrich). The tissue sections were blocked for 30 min in 10% normal goat serum and 2% BSA in PBS. A 5 h incubation period with the primary antibody was followed by a 30 min incubation with biotinylated goat anti-rabbit IgG (1:200 dilution). Subsequently, sequential incubations with secondary antibody blocker, streptavidin-HRP, and Tyramide Alexa Fluor 568 (Invitrogen) were performed. Confocal images were obtained using laser scanning confocal microscopy (Olympus, Fukuoka, Japan).

### 4.8. Dual-Luciferase Reporter Assay

Luciferase activity was determined by using the dual-luciferase reporter assay system (Promega, Madison, WI, USA). The promoter region of the SerRS gene, and that of the RARE1 deletion (−845 to −831) or RARE2 deletion (−2333 to −2318), were amplified by PCR and cloned into the pGL4.11 [luc2P] vector (between Kpn I and Xho I sites) to create the pGL4–SerRS–WT, pGL4–SerRS–△RARE1, and pGL4–SerRS–△RARE2 firefly luciferase reporter plasmids. After 16 h of incubation in 12-well plates, HaCaT cells were transiently transfected with 100 ng of the aforementioned reporter plasmids, and 500 ng of the pFlag–CMV-2 empty vector was transfected as the control. A Renilla luciferase control reporter plasmid pRL-SV40 (50 ng) was co-transfected to normalize the transfection efficiency among different experiments. Six hours after transfection, the cells were treated with DMSO or *t*RA (3 μM), respectively. Dual-luciferase reporter assays were performed 12 h after treatment.

## 5. Conclusions

In summary, this study has revealed a novel signaling cascade that controls VEGFA expression upon UV exposure and is mediated by the ATM–SerRS pathway. Moreover, we discovered the undetected mechanism of *t*RA in inhibiting UV-induced VEGFA expression and angiogenesis. Taken together with this study, our findings contribute to a new understanding of UV irradiation on human skin and provide a promising approach for the prevention of angiogenesis-related skin damage and melanoma progression by targeting ATM and SerRS.

## Figures and Tables

**Figure 1 cancers-11-01847-f001:**
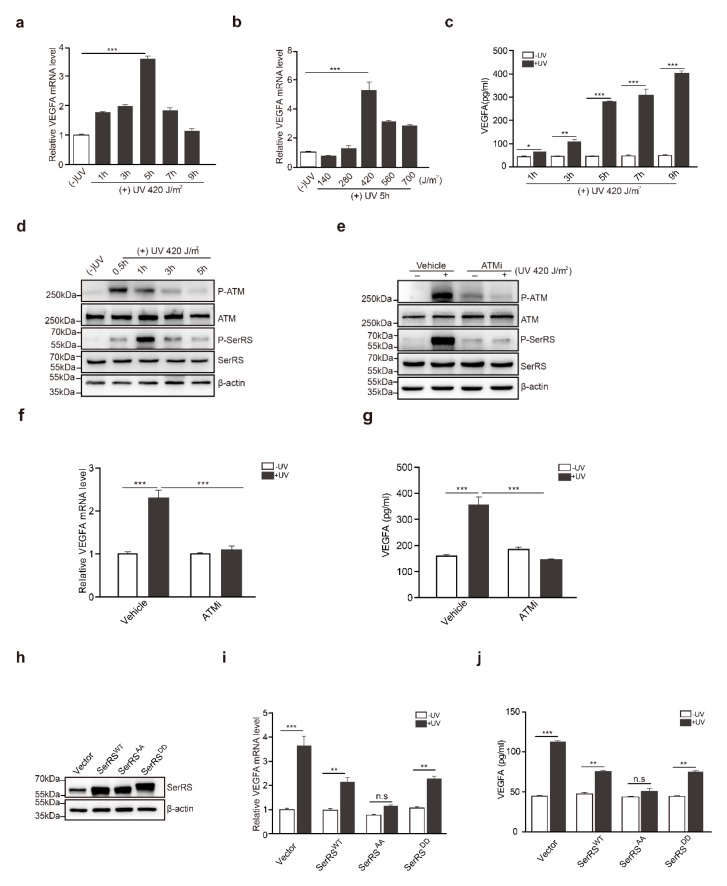
The ataxia telangiectasia mutated (ATM)–seryl-tRNA synthetase (SerRS)–vascular endothelial growth factor A (VEGFA) pathway played an essential role in UV-induced VEGFA expression in HaCaT cells. (**a**,**b**) mRNA levels of *VEGFA* were analyzed by RT-PCR. (**c**) VEGFA in supernatants was analyzed via sandwich ELISA. (**d**) Western blot analysis of P-ATM, ATM, P-SerRS, and SerRS expressions in HaCaT cells with or without UV treatment. (**e**) Western blot analysis of P-ATM, ATM, P-SerRS, and SerRS expressions in HaCaT Cells with or without treatments of UV and ATMi. (**f**) VEGFA mRNA levels were detected by RT-PCR. (**g**) VEGFA in supernatants was analyzed via sandwich ELISA. (**h**) Western blot analysis of SerRS expression; β-actin served as a loading control. (**i**) mRNA levels of *VEGFA* were analyzed by RT-PCR. (**j**) VEGFA in supernatants was analyzed via sandwich ELISA. All data above are presented as means ± SEM (*n* = 3, * *p* < 0.05, ** *p* < 0.01, *** *p* < 0.001) of three independent repeats.

**Figure 2 cancers-11-01847-f002:**
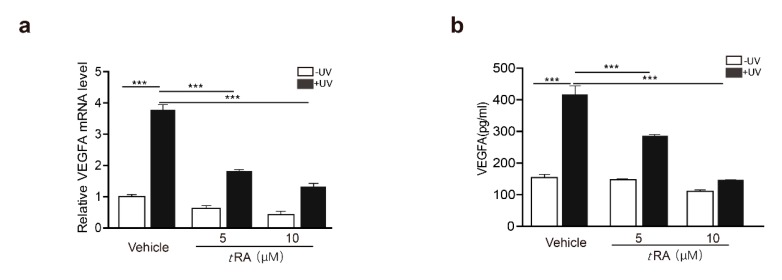
All-trans retinoic acid (*t*RA) enhanced the transcription of SerRS to suppress VEGFA expression in HaCaT cells. HaCaT cells were pretreated with vehicle (DMSO) or *t*RA 24 hours before UV radiation. (**a**) mRNA levels of *VEGFA* in HaCaT cells were detected by RT-PCR. (**b**) VEGFA secretion was determined via ELISA. (**c**) Details of constructed plasmids, all with luciferase reporter, but discriminatively with intact SerRS promoter or the deleted binding sites of RARE1 or RARE2. (**d**) Relative SerRS activity was analyzed via luciferase assay in HaCaT-SerRS^WT^ cells. (**e**) Relative SerRS activity was analyzed via luciferase assay in HaCaT-SerRS^WT^, HaCaT-SerRS^ΔRARE1^, and HaCaT-SerRS^ΔRARE2^ cells. (**f**) Transcriptional levels of SerRS were analyzed with RT-PCR. (**g**) Western blot analysis of SerRS expression; β-actin serves as a loading control. All values above are shown as means ± SEM (*n* = 3, ** *p* < 0.01, *** *p* < 0.001).

**Figure 3 cancers-11-01847-f003:**
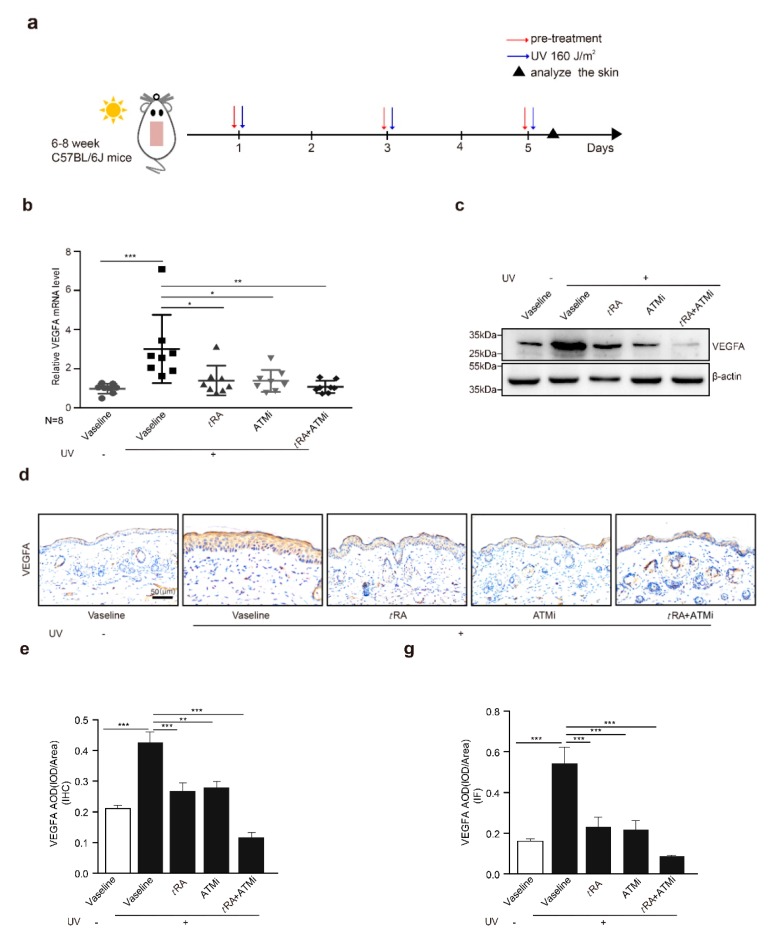
ATM inhibitor enhanced the effect of *t*RA on the protection of mouse skin from UV-induced VEGFA expression. (**a**) Experimental model. (**b**) mRNA of *VEGFA* was quantified by RT-PCR. (**c**) Western blot was adopted to analyze VEGFA expression. (**d**) Representative images of VEGFA immunohistochemical staining in cutaneous tissues from C57BL/6 mice with indicated treatments. (**e**) Quantification of VEGFA immunohistochemical staining. (**f**) Immunofluorescence analysis for VEGFA in the skin of mice with indicated treatments. (**g**) Quantification of VEGFA immunofluorescence analysis. Three different fields per section were examined. Data are presented as means ± SEM for three subjects. (*n* = 8, * *p* < 0.05, ** *p* < 0.01, and *** *p* < 0.001 versus control group by the Wilcoxon signed-rank test).

**Figure 4 cancers-11-01847-f004:**
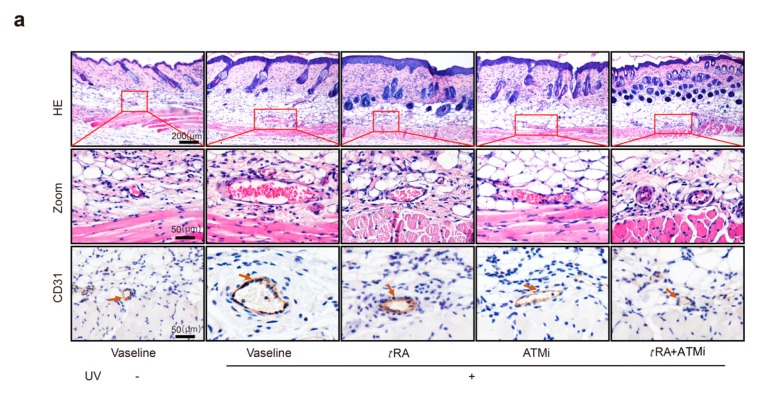
ATM inhibitor improved the effect of *t*RA against pathological skin angiogenesis caused by UV irradiation in mice. (**a**) Representative H&E (upper panel) and platelet/endothelial cell adhesion molecule-1 (CD31)-immunostained (lower panel) skin sections of C57BL/6 mice exposed to indicated treatments. (**b**) Vessel size, (**c**) vessel density, and (**d**) blood vessel area within 200 μm from the epidermal–dermal junction were analyzed with a computer-assisted morphometric analysis program. Three different fields per section were examined. Data are presented as means ± SEM for three subjects. *** *p* < 0.001 versus control group (Wilcoxon signed-rank test).

**Figure 5 cancers-11-01847-f005:**
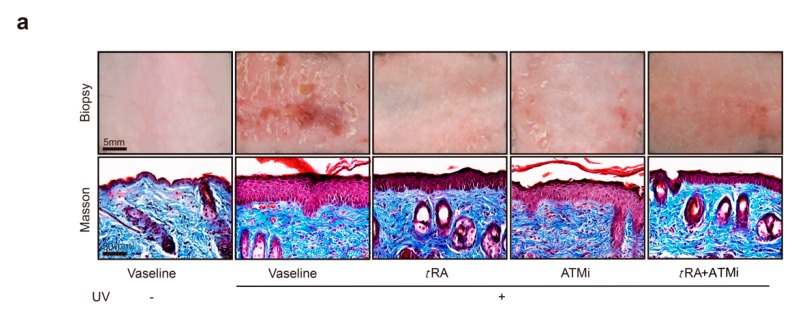
*t*RA and ATMi alleviated thickening and the inflammatory response of mice skin exposed to UV. (**a**) Representative bare skin (upper panel) and Masson-stained skin sections (lower panel) of C57BL/6 mice exposed to indicated treatments. (**b**) Analysis of epidermal thickness; each section was measured three times at different positions. Data are presented as means ± SEM. (**c**) Quantification of skin response to UV radiation with the LSR (local skin response) grading scale. (**d**) mRNA levels of IL-1b were determined via RT-PCR. (**e**) mRNA levels of IL-10 were determined via RT-PCR. (**f**) mRNA levels of TNF-α were determined via RT-PCR. (**g**) mRNA levels of INF-α were determined via RT-PCR. Data are presented as means ± SEM (*n* = 8, ** *p* < 0.01, *** *p* < 0.001).

**Figure 6 cancers-11-01847-f006:**
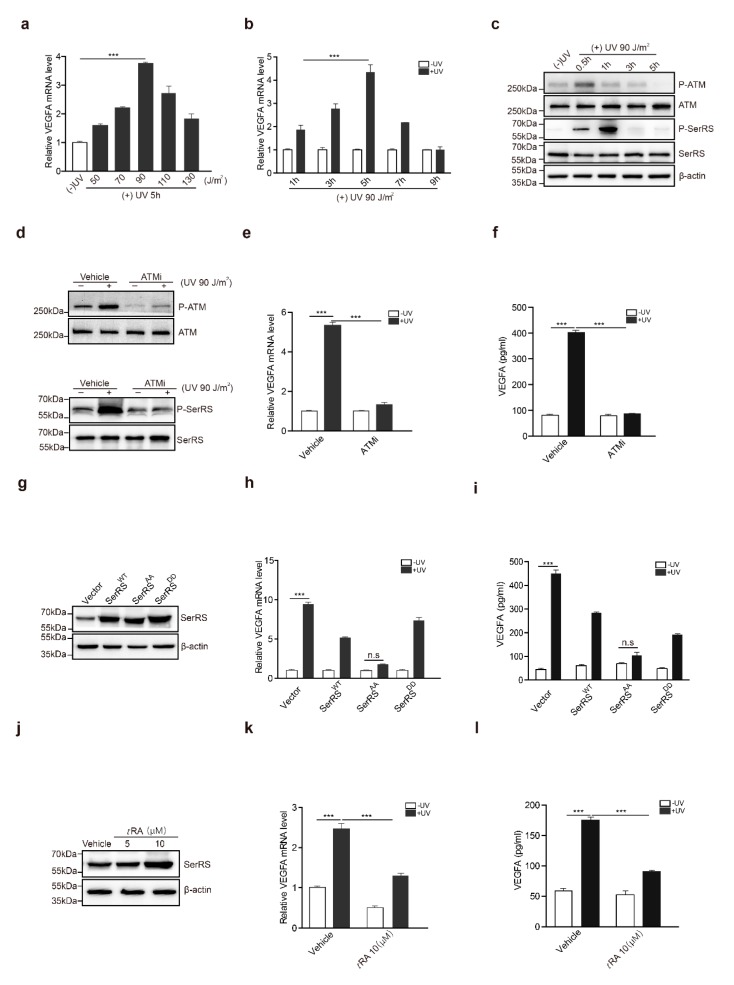
*t*RA and ATMi inhibited UV-induced VEGFA expression in melanoma cells. (**a**) mRNA levels of *VEGFA* in B16F10 cells with or without UV radiation were analyzed by RT-PCR. (**b**) mRNA levels of *VEGFA* in B16F10 cells harvested at indicated time-points were analyzed by RT-PCR. (**c**) Western blot analysis of P-ATM, ATM, P-SerRS, and SerRS expressions in B16F10 cells with or without UV treatment. (**d**) Western blot analysis of P-ATM, ATM, P-SerRS, and SerRS expressions in B16F10 cells with or without UV and KU-55933 (10μM) treatments. (**e**) *VEGFA* mRNA levels were detected by RT-PCR. (**f**) VEGFA in supernatants was analyzed via sandwich ELISA. (**g**) Western blot analysis of SerRS expression in B16F10 cells. (**h**) *VEGFA* mRNA levels were analyzed via RT-PCR. (i) VEGFA in supernatants was analyzed via sandwich ELISA. (**j**) Western blot analysis of SerRS expression; β-actin served as a loading control. (**k**) mRNA levels of *VEGFA* were analyzed by RT-PCR. (**l**) VEGFA in supernatants was analyzed via sandwich ELISA. All data above are presented as means ± SEM (*n* = 3, * *p* < 0.05, ** *p* < 0.01, *** *p* < 0.001) of three independent repeats.

**Figure 7 cancers-11-01847-f007:**
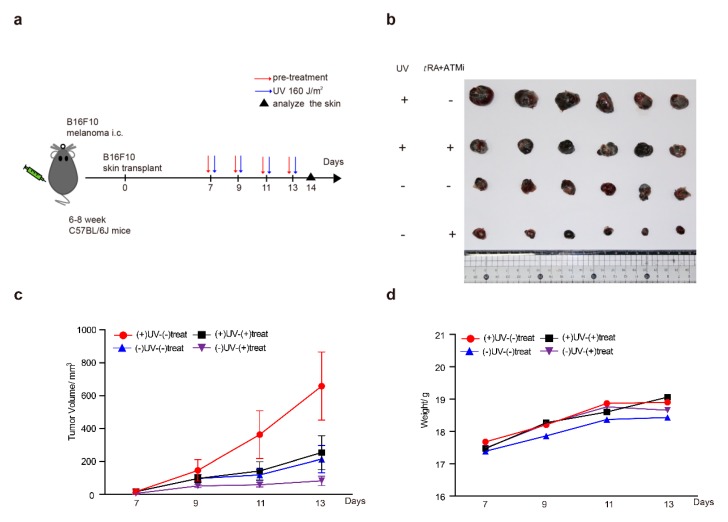
ATM inhibitor greatly improved the effect of *t*RA to inhibit the growth of a melanoma xenograft in mice. (**a**) Schematic diagram of the experimental procedure for the xenograft model in C57BL/6 mice. (**b**) Representative tumor images are shown. (**c**) Tumor growth and (**d**) body weight curves of the mice treated with UV and *t*RA/KU-55933 and the controls. (**e**) Western blot analysis of VEGFA expression in tumor tissues from mice treated with UV and *t*RA/KU-55933 and the controls. (**f**) RT-PCR analysis of VEGFA mRNA levels in tumor tissues from the mice mentioned above. All values above are shown as means ± SEM (*n* = 8, ** *p* < 0.01).

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
