# Peer review of "Targeting Angiogenesis by Blocking the ATM–SerRS–VEGFA Pathway for UV-Induced Skin Photodamage and Melanoma Growth"

_cancers, 2019, doi:10.3390/cancers11121847_

Round 1

Reviewer 1 Report

In this study Song et al describe the role of blocking ATM-SerRs-VEGFA pathway for treating skin photodamage induced by UV and inhibiting melanoma growth. 

The experiments are well designed and executed. However, more effort is needed to descript and discuss the results of this work.

My feeling is that the work described here gives important proof of principle data. As such I have only some suggestions to improve the manuscript.

The authors need to consider simplifying the language and try to use short sentences.

The results part is a mix between of result and M&M. therefore, I would suggest to move all paragraphs which describing the experiments to the M&M. The same is true for figures’ legends. Here the authors need to describe and explain the figures and not how they did the experiments.

The authors need to expand the discussion to reflect their findings.

The authors are mention their unpublished data several times through the manuscript (in the introduction, results and discussion) and describe it as “in publication”. I do not think this is the wright way to include this data in the publication. I would suggest not to consider unpublished data too often specially in the discussion part.

Figure 1a shows different time point for UV treatments and the legend refer that the cells were harvested after 9 h. this is confusing and need to be explain.

The order of sub-figures and their presence in the legend text need to be identical. For i.e. in figure 1i was described before figure 1 g and h.

The authors need to consider simplifying the language and try to use short sentences.

Please expand the acronyms for, i.e. what is ATM.

Author Response

Response to Reviewer 1 Comments

Point 1: The authors need to consider simplifying the language and try to use short sentences.

Response 1: The MDPI English Editing Service has helped us edit the manuscript carefully, and improved the language significantly.

Point 2: The results part is a mix between of result and M&M. therefore, I would suggest to move all paragraphs which describing the experiments to the M&M. The same is true for figures’ legends. Here the authors need to describe and explain the figures and not how they did the experiments.

Response 2: Thank the reviewer for pointing this out. We have re-edited the manuscript and corrected the deficiencies, and also moved all the sentences describing how we did the experiments from Results and Figures’ legends to the M&M. We have simplified and standardized the descriptions in Figures’ legends based on the suggestion.  

Point 3: The authors need to expand the discussion to reflect their findings.

Response 3: We expanded the discussion from the following two aspects:

(1) In consideration of the limited therapeutic effect of present anti-VEGF drugs for the only mechanism directly neutralizing mature VEGF in serum, we discussed the significance of our results as follows: “Almost all the anti-VEGF therapeutic approaches present are monoclonal antibodies directly targeting one or more mature isoforms of VEGF in serum, or VEGF receptors, thereby inhibiting tumor growth [1,2]. Such limited mechanisms result in an urgent issue to be understood that why the majority of patients do not respond or stop responding to such drugs. Therefore, our study on the regulation of VEGFA transcription, the source of this important growth factor, is innovative and important”.

(2) Immunotherapy with PD1/PD-L1 inhibitors has demonstrated amazing clinical benefit in melanoma, but with frequent neurotoxicity and cardiotoxicity, hence we discussed the possibility and prospect of the combined application of tRA and ATM inhibitor with immunotherapy: “Immunotherapy with PD-L1/PD-1 inhibitors has demonstrated clinical benefit across a wide range of cancer types, especially malignant melanoma. However, patients who experience durable response and survival are only a limit subset of those treated. Therefore, we need to identify novel combinations. In fact, the biologic roles of VEGFA have extended beyond its impact on angiogenesis over the years. It can have a direct effect on multiple cells involved in immunity, including T cells, dendritic cells, regulatory T cells and myeloid-derived suppressor cells. Recent studies have also showed that anti-VEGF therapy can improve anti-PD-L1 (programmed death ligand 1) treatment, especially when it generates intratumoral high endothelial venules (HEVs) that facilitate enhanced cytotoxic T lymphocyte (CTL) infiltration and tumor cell destruction [3]. Recent clinical studies also support the role of VEGFA in anti-cancer immunity [4]. Though it is tempting to speculate the anti-vascular effects of VEGF inhibition in combination with immunotherapy, the mechanism of the anti-VEGF/anti-PD-L1 interaction is multifactorial. The addition of tRA and ATM inhibitor may increase such toxicity further. However, to overcome these difficulties is still promising and anticipated”.

Point 4: The authors are mention their unpublished data several times through the manuscript (in the introduction, results and discussion) and describe it as “in publication”. I do not think this is the wright way to include this data in the publication. I would suggest not to consider unpublished data too often especially in the discussion part.

Response 4: Thank the reviewer for raising this out. We have revised the description.

Point 5: Figure 1a shows different time point for UV treatments and the legend refer that the cells were harvested after 9 h. this is confusing and need to be explain.

Response 5: We apologize for the mistaken description and we have amended it. Cells were harvested at indicated time points after UV irradiation, and the last time is at 9 hours after UV.

Point 6: The order of sub-figures and their presence in the legend text need to be identical. For i.e. in figure 1i was described before figure 1g and h.

Response 6: We have corrected the mistakes and rechecked the order of all the sub-figures and their presence in the legend text.

Point 7: The authors need to consider simplifying the language and try to use short sentences.

Response 7: The MDPI English Editing Service has helped us edit the manuscript carefully, significantly improved the language.

Point 8: Please expand the acronyms for, i.e. what is ATM.

Response 8: Thank the reviewer for pointing this out. We have checked and added in all the definitions where needed.

Ebos, J.M.; Kerbel, R.S. Antiangiogenic therapy: impact on invasion, disease progression, and metastasis. Nature reviews. Clinical oncology 2011, 8, 210-221, doi:10.1038/nrclinonc.2011.21. Apte, R.S.; Chen, D.S.; Ferrara, N. VEGF in Signaling and Disease: Beyond Discovery and Development. Cell 2019, 176, 1248-1264, doi:10.1016/j.cell.2019.01.021. Allen, E.; Jabouille, A.; Rivera, L.B.; Lodewijckx, I.; Missiaen, R.; Steri, V.; Feyen, K.; Tawney, J.; Hanahan, D.; Michael, I.P., et al. Combined antiangiogenic and anti-PD-L1 therapy stimulates tumor immunity through HEV formation. Science translational medicine 2017, 9, doi:10.1126/scitranslmed.aak9679. Kelly, P.N. The Cancer Immunotherapy Revolution. Science (New York, N.Y.) 2018, 359, 1344-1345, doi:10.1126/science.359.6382.1344.

Reviewer 2 Report

This study investigated the all-trans retinoic acid (tRA) can induce the expression of a newly identified potent anti-angiogenic factor seryl tRNA synthetase. tRA could activate the transcription of seryl tRNA synthetase through the binding with seryl tRNA synthetase promoter. tRA combined with ATM inhibitor KU-55933, the inhibition of VEGFA expression was increased as well as the protection of mouse skin from photodamage. The combination greatly inhibited tumor angiogenesis and growth in mouse melanoma xenograft in vivo.

This study was interesting but some points must be clarified.

The UV dose was 420 J/m2 in this study. Although the authors test the VEGFA expression after 140-700 J/m2 UV exposure, the high dose of UV exposure may cause cell death. The reason for used really high dose of UV and the results of cell viability and morphology of the experiment must present in the manuscript. In addition, the UV dose used in animal study is 160 J/m2. It is unreasonable that the dose in cell model was higher than animal study. This work studied the effect of UV on VEGFA expression in HaCaT and melanoma cells, however, the effect and mechanism of UV on VEGFA expression and photodamage and photocarcinogenesis was unclear. More experiments were needed. In this study, the application of tRA alone did not fully inhibit UV-induced VEGFA and the results were shown in Figure 2. How about the dose of tRA? Authors stated that “Given that the application of tRA alone did not fully inhibit UV-induced VEGFA (Fig. 2a and 2b), while overexpression of phosphorylation-deficient SerRS mutant could almost fully repress UV-induced VEGFA (Fig. 1i and 1j), we suspected that the inefficiency of tRA might be due to SerRS phosphorylation by UV-activated ATM, which inhibited the transcriptional repressor activity of SerRS.” And “To test, we treated mice with tRA alone or tRA combined with ATM inhibitor KU-55933 before acute UV irradiation on the mouse skin (Fig. 3a).” Why the study did not perform in cell model but the animal model? The abbreviations have to define at the first shown in the manuscript. For example ATM. There some typing and grammar errors, please check the text carefully.

Round 2

Reviewer 2 Report

The authors have done several experiments and revised their manuscript accordingly.